# Apo-Lactoferrin Inhibits the Proteolytic Activity of the 110 kDa Zn Metalloprotease Produced by *Mannheimia haemolytica* A2

**DOI:** 10.3390/ijms25158232

**Published:** 2024-07-28

**Authors:** Gerardo Ramírez-Rico, Lucero Ruiz-Mazón, Magda Reyes-López, Lina Rivillas Acevedo, Jesús Serrano-Luna, Mireya de la Garza

**Affiliations:** 1Facultad de Estudios Superiores Cuautitlán, Universidad Nacional Autónoma de México, Estado de México 54714, Mexico; garmvz@gmail.com; 2Departamento de Biología Celular, Centro de Investigación y de Estudios Avanzados del Instituto Politécnico Nacional, Ciudad de México 07360, Mexico; luceroruizmazon@gmail.com (L.R.-M.); magda.magrel2003@gmail.com (M.R.-L.); jesus.serrano@cinvestav.mx (J.S.-L.); 3Centro de Investigación en Dinámica Celular, Instituto de Investigación en Ciencias Básicas y Aplicadas, Universidad Autónoma del Estado de Morelos, Cuernavaca 62209, Mexico; Irivillas@uaem.mx

**Keywords:** *Mannheimia haemolytica*, 110-Mh metalloprotease, proteases, zymography, bovine apo-lactoferrin

## Abstract

*Mannheimia haemolytica* is the main etiological bacterial agent in ruminant respiratory disease. *M. haemolytica* secretes leukotoxin, lipopolysaccharides, and proteases, which may be targeted to treat infections. We recently reported the purification and in vivo detection of a 110 kDa Zn metalloprotease with collagenase activity (110-Mh metalloprotease) in a sheep with mannheimiosis, and this protease may be an important virulence factor. Due to the increase in the number of multidrug-resistant strains of *M. haemolytica*, new alternatives to antibiotics are being explored; one option is lactoferrin (Lf), which is a multifunctional iron-binding glycoprotein from the innate immune system of mammals. Bovine apo-lactoferrin (apo-bLf) possesses many properties, and its bactericidal and bacteriostatic effects have been highlighted. The present study was conducted to investigate whether apo-bLf inhibits the secretion and proteolytic activity of the 110-Mh metalloprotease. This enzyme was purified and sublethal doses of apo-bLf were added to cultures of *M. haemolytica* or co-incubated with the 110-Mh metalloprotease. The collagenase activity was evaluated using zymography and azocoll assays. Our results showed that apo-bLf inhibited the secretion and activity of the 110-Mh metalloprotease. Molecular docking and overlay assays showed that apo-bLf bound near the active site of the 110-Mh metalloprotease, which affected its enzymatic activity.

## 1. Introduction

*Mannheimia haemolytica* is a Gram-negative opportunistic bacterium that resides as a commensal organism in the nasopharynx and tonsil regions of the conduction system of the respiratory tract of healthy ruminants [1]. A primary viral infection and concomitant stressors that decrease immunity cause secondary colonization by *M. haemolytica* in the exchange system of the respiratory system [2]. Clinically, pneumonic mannheimiosis is characterized by severe toxemia that can kill animals even when considerable parts of the lungs remain functionally and structurally normal. Cattle generally become depressed, febrile (40 °C to 41 °C), and anorexic and suffer from a productive cough, encrusted nose, mucopurulent nasal exudate, shallow respiration, or an expiratory grunt [3]. The result of colonization by *M. haemolytica* in the exchange system is fibrinopurulent pleurobronchopneumonia. Due to the necrotizing process, sequelae to pneumonic mannheimiosis can be serious and include abscesses, encapsulated sequestra (isolated pieces of necrotic lung), chronic pleuritis, fibrous pleural adhesions, and bronchiectasis [4].

In recent years, *M. haemolytica* A2 isolates from sheep with mannheimiosis have revealed an increase in antimicrobial resistance. The most common resistance is to beta-lactams, tetracyclines, sulfonamides, and aminoglycosides [5,6]. Due to the lack of a vaccine with 100% efficacy, new strategies must be developed to reduce the presence of this disease on farms. Lactoferrin (Lf) may be an option as a preventive or therapeutic agent for *M. haemolytica*. We reported that apo-bLf has a bactericidal and bacteriostatic effect towards *M. haemolytica* A2 [7,8].

Lactoferrin, an iron-binding multifunctional glycoprotein that is highly conserved among mammals, is worthy of being called a “miracle molecule” [9]. A number of authors have reported the bacteriostatic and bactericidal effects of apo-bovine lactoferrin (apo-bLf) (the iron-free molecule) on the growth of important human and veterinary pathogens. Lactoferrin exhibits several other beneficial properties, such as anticancer, anti-inflammatory, immunomodulatory, and DNA-regulatory activities. Recent reports have shown the therapeutic properties of lactoferrin in the treatment of neurodegenerative diseases associated with aging and stress-related emotional disorders [9,10,11]. Our research group reported that the addition of subinhibitory concentrations of apo-bLf to the culture medium of *M. haemolytica* A2 affected the secretion of proteolytic activities [12]. However, the effects of apo-bLf on purified or already secreted proteases in this bacterium have not been studied. We recently identified a 110 kDa Zn metalloprotease with collagenase activity (110-Mh metalloprotease) in necrotic lesions caused by *M. haemolytica* A2 [13]. This protease may contribute significantly to the pathogenesis of pneumonic mannheimiosis and the inflammatory response. It has been reported that collagenases participate in necrotic processes, act as virulence factors for invasiveness and tissue injury, and promote the inflammatory process [14,15]. Therefore, in this study, our objective was to inhibit the secretion and activity of the 110-Mh metalloprotease of *M. haemolytica* A2 with apo-bLf and determine its possible mechanism of action. We observed that apo-bLf inhibited the activity of the 110-Mh metalloprotease using enzymatic kinetics and zymography and azocoll assays. Molecular docking and overlay showed that apo-bLf bound to the 110-Mh metalloprotease primarily via hydrophobic bonds close to the active site, which caused proteolytic inhibition. Therefore, inhibition of the 110-Mh metalloprotease could reduce necrotic tissue and the inflammation caused by *M. haemolytica*, which are important pathological disorders responsible for death in mannheimiosis. These results suggest that apo-bLf, primarily in combination with antimicrobials, may be used in the treatment of this bacterium.

## 2. Results

### 2.1. Effect of Apo-bLf on the Growth of M. haemolytica A2

To determine the subinhibitory concentrations of apo-bLf for obtaining culture supernatants (SNs) and to determine the effect of apo-bLf on proteolytic activity, growth kinetics were performed with final concentrations of 5, 6, and 8 µM apo-bLf for 24 h at 37 °C with agitation (200 rpm). As shown in Figure 1, the only concentration of apo-bLf that exhibited a bactericidal effect was 8 µM, and 5 and 6 µM had no effect on the growth of *M. haemolytica* A2. Therefore, these concentrations were considered subinhibitory and used in the subsequent assays.

### 2.2. Bovine Apo-Lactoferrin Inhibits the Secretion of M. haemolytica A2 Proteases, as Detected with an Anti-110 kDa Synthetic Peptide Antibody

We used the subinhibitory concentrations of apo-bLf to evaluate the proteolytic effect of *M. haemolytica* A2 using zymography. Here, 5 and 6 µM apo-bLf (Figure 2a, lanes 1 and 2, respectively) inhibited the proteolytic activity compared to the degradation control (Figure 2a, lane 3). The protease activity and the inhibition of its secretion were confirmed by Western blotting after the generation of an antibody using a synthetic peptide. For this purpose, a sequence near the active site was chosen (residues 104–127) on the basis of hydrophobicity, immunogenicity, accessibility, and polarity. Only the 110-Mh metalloprotease was detected when *M. haemolytica* was grown without apo-bLf (Figure 2b, lane 3). These results confirmed the identification of the 110-Mh metalloprotease and the inhibition of its secretion by apo-bLf.

### 2.3. Bovine Apo-Lactoferrin Inhibits the Collagenase Activity of the 110-Mh Metalloprotease of M. haemolytica A2

Our previous results demonstrated that subinhibitory concentrations of apo-bLf inhibited proteolytic activity when the bacteria were incubated with *M. haemolytica* A2 culture media. Subsequent assays evaluated the effect of apo-bLf on the previously purified 110-Mh metalloprotease. Zymograms were copolymerized with collagen, and collagen degradation kinetics were determined using SDS-PAGE. Incubation of 110-Mh metalloprotease with 5 or 6 µM apo-bLf drastically inhibited the collagenolytic activity (Figure 3a, Lanes 2 and 3). The inhibition of 110-Mh metalloprotease activity was also evaluated using SDS-PAGE. This assay did not reveal the formation of peptides derived from collagen chains when 5 and 6 µM apo-bLf were incubated with 110-Mh metalloprotease (Figure 3b, lanes 2 and 3) compared to the 110-Mh metalloprotease without apo-bLf and incubation with collagen (Figure 3b, lane 4). These results demonstrated that apo-bLf inhibited the collagenolytic activity of the 110-Mh metalloprotease. Lactoferrin may interact with the active site of the 110-Mh metalloprotease and counteract its activity.

### 2.4. Inhibition Assays of the 110-Mh Metalloprotease Activity with Apo-bLf Using Azocoll

After zymography and SDS-PAGE, collagenolytic activity inhibition assays on azocoll were performed. The 110-Mh metalloprotease was incubated with the same concentrations of apo-bLf as above and compared with proteins from *M. haemolytica* A2 SNs preincubated in culture medium with apo-bLf. The 110-Mh metalloprotease without apo-bLf and 110-Mh metalloprotease incubated with phenanthroline (a specific inhibitor of zinc-dependent metalloproteases) were used as experimental controls. A chromogenic assay was subsequently performed with azocoll. The results macroscopically revealed an inhibition of azocoll degradation in all of the assays with apo-bLf, with a lighter color than the 110-Mh metalloprotease without apo-bLf (Figure 4a). These results were quantified and showed statistical significance in assays where apo-bLf was used compared to assays of the 110-Mh metalloprotease without bLf (Figure 4b).

### 2.5. Bovine Lactoferrin Binds with High Affinity to the 110-Mh Metalloprotease, as Determined by In Silico Analysis

As the binding of apo-bLf to the 110-Mh metalloprotease may be the mechanism of its inhibition of proteolytic activity, we used in silico analysis to evaluate this hypothesis. Molecular docking was performed between bLf and the 110-Mh metalloprotease. The results showed the 3D structure of the 110-Mh metalloprotease obtained by molecular modeling from its primary amino acid sequence and its coupling with bLf. The coupling energy values, given in Gibbs free energy, indicated great affinity between both molecules. This interaction primarily occurred via hydrophobic interactions that were near the active site of the 110-Mh metalloprotease (Figure 5). These data suggest that the binding of bLf with the 110-Mh metalloprotease produces a steric effect on the protease and inhibits its activity.

### 2.6. Binding of Bovine Apo-Lf to the 110-Mh Metalloprotease, as Determined by Overlay Blot

After determining that apo-bLf interacted with the 110-Mh metalloprotease via molecular docking analysis, we evaluated its interaction in vitro using overlay assays. We performed electrophoretic separation and electrotransferred the 110-Mh metalloprotease to a nitrocellulose membrane. Next, we incubated it with apo-bLf and detected the reaction using anti-bLf antibodies. The results confirmed the findings of the docking analysis as a main band of 110 kDa was found (Figure 6b). Both results (in silico and in vitro, Figure 5 and Figure 6, respectively) demonstrated that apo-bLf bound to the 110-Mh metalloprotease, which may be one of the mechanisms of proteolytic inhibition.

## 3. Discussion

*Mannheimia haemolytica* is a Gram-negative bacterium and the primary species associated with ruminant respiratory disease, which is a multifactorial condition in cattle and sheep that involves poorly understood interactions between various bacterial and viral pathogens and the host [16]. Antibiotics play a crucial role in the control of mannheimiosis. However, the emergence of multidrug-resistant bacterial strains is increasing, which raises serious public health concerns about the increased use of antimicrobial drugs in milk- and food-producing animals [17]. Therefore, alternative strategies that rely less extensively on drugs are needed to control this disease. Lactoferrin is a multifunctional miracle molecule that is known for its numerous antibacterial effects [18,19]. We began our study by assessing the impact of apo-bLf on the growth of *M. haemolytica* A2. Our findings indicated that concentrations of 5 µM and 6 µM apo-bLf were subinhibitory for *M. haemolytica* A2. Our results are consistent with studies in 2020 and 2021, in which subinhibitory concentrations of 1 µM and 3.5 µM were used to evaluate the virulence factors of *M. haemolytica.* In addition, viability assays were performed over a period of 9 h, which identified bactericidal concentrations of apo-bLf at 7.5, 8, 16, and 20 µM [7,12]. However, the present study used relatively high subinhibitory concentrations of apo-bLf (5 and 6 µM) and an incubation time of 24 h because this time was needed to obtain the 110-Mh metalloprotease. Despite the use of extended incubation times, apo-bLf had bactericidal effects even at a low concentration (8 µM). The antimicrobial activity of Lf is related to its ability to bind iron, which inhibits bacterial growth by depriving bacteria of this essential nutrient via its ability to act as a bacteriostatic agent. Lf has antimicrobial activity via iron-independent pathways and direct interactions with membrane components [20]. Samaniego et al. reported that apo-bLf bound to a porin of 34.2 kDa and an outer membrane protein (OmpA) of 32.9 kDa of *M. haemolytica* A1, which is a cattle pathogen, and caused membrane destabilization and subsequent lysis [8].

An important pathogenic mechanism studied in recent years is the secretion of proteases that degrade host proteins and tissues. These enzymes help pathogens evade the immune response and promote necrotic lesions, inflammation, and colonization in target tissues [21,22]. A 35 kDa metallo-sialoglycoprotease that degrades sialoglycoproteins of leukocytes and the lung epithelium has been described in all serotypes of *M. haemolytica,* except for serotype A11 and biotype T [23]. Rosales-Islas et al. (2024) reported the isolation and characterization of a 70 kDa serine protease from *M. haemolytica* A1 that specifically targets ovine and bovine fibrinogen, which is an important host protein that accumulates in the alveoli and pleura, marking a critical finding in mannheimiosis [24]. We studied the proteolytic activities of *M. haemolytica* serotype A2, which is primarily an ovine pathogen, in SNs and outer membrane vesicles released by the bacterium. We identified primarily nonspecific cysteine and metalloproteases of high molecular weight (250 kDa) that degraded fibrinogen, hemoglobin, albumin, apo- and holo-bLf, and collagen [25]. We recently performed in vitro purification and in vivo detection of a 110 kDa substrate-specific Zn metalloprotease that targeted bovine type I collagen. We identified this 110-Mh metalloprotease predominantly in necrotic lesions and inflammatory exudates, which suggests that it plays a significant antigenic role in the pathogenesis of *M. haemolytica* A2 [13].

Therefore, in this study, we focused on investigating the effect of apo-bLf as an inhibitor of 110-Mh metalloprotease activity. Our results revealed that subinhibitory concentrations of apo-bLf inhibited the secretion and activity of the 110-Mh metalloprotease, as demonstrated by Western blot and zymography analyses. Similarly, azocoll assays confirmed that apo-bLf inhibited the proteolytic activity of the 110 kDa collagenase, with degradation levels similar to the inhibition observed when the 110-Mh metalloprotease was incubated with phenanthroline, a known zinc metalloprotease inhibitor. Similar results were reported by Nakayama et al. in 2008. They demonstrated that recombinant human lactoferrin (r-hLf) inhibited matrix metalloproteinase activity (MMP-2, MMP-3, and MMP-9) in a rabbit preterm delivery model. These metalloproteases are used as markers of cervical maturation [26]. The inhibition of MMP activity by r-hLf may be due to two main mechanisms: binding to the catalytic site of the metalloprotease and/or chelation of its prosthetic group—in this case, the zinc ion. Based on these findings, we next analyzed whether apo-bLf bound to the 110-Mh metalloprotease. In silico analysis revealed a high affinity between apo-bLf and the 110-Mh metalloprotease, primarily due to hydrophobic interactions near the collagenase catalytic site. This interaction was subsequently confirmed using overlay tests. We proposed that this binding induced an allosteric effect on the protease, which led to its inhibition. These findings are consistent with Zwirzitz et al. (2018). They showed that r-hLf inhibited plasminogen activation by directly binding to human plasminogen, which indicated that h-Lf played a role in circulatory disorders, particularly the hemostasis process [27].

However, the possibility that apo-bLf also exerts a chelating effect on the zinc ions in the 110-Mh metalloprotease cannot be excluded. Despite its high affinity for iron, Lf binds to various other divalent metal cations, including Zn^2^⁺, Mg^2^⁺, and Co^2^⁺, albeit with a lower affinity than for iron [28]. Complexes of Lf with Zn^2^⁺ have demonstrated significant antimicrobial and antiviral activities against pathogens [29,30,31]. In this context, Newsome et al. (2007) reported that hLf removed the catalytic zinc from the active site of MMP-2. To determine the inhibition mechanism, increasing concentrations of ZnCl_2_ were added to the reaction, which resulted in the recovery of MMP-2 activity. Zinc-saturated Lf did not inhibit MMP-2, which confirmed that hLf acted as a reversible inhibitor of metalloproteases by chelating zinc ions [32]. 

The effect of Lf on proteases has been demonstrated in other Gram-negative bacteria, and numerous studies highlight proteases as critical virulence factors. Luna-Castro et al. used zymography assays and reported that apo-bLf inhibited metalloproteases secreted by *Actinobacillus pleuropneumoniae* serotype 1 [33]. Using immunoblot assays, Gomez et al. reported that apo-bLf inactivated IgA protease 1 and the HAP adhesin of *Haemophilus influenzae* [34]. Trials with anti-sialoglycoprotease vaccines from *M. haemolytica* A1 revealed a reduction in lung lesions at calf necropsy compared to the use of anti-LKT antibodies, which showed no effect. Studies with *Biberstenia trehalosi* demonstrated that the use of antiprotease serum in sheep resulted in less necrotic lung tissue compared to the control group. These findings demonstrate that bacterial proteases may be the primary pathogenic determinants responsible for necrotic or exudative lesions caused by pathogenic bacteria. It would be interesting to evaluate in future studies whether apo-bLf, in combination with an antimicrobial, could reduce necrotic lesions and inflammatory infiltrate in animals with pneumonic mannheimiosis. Furthermore, apo-bLf, due to its immunomodulatory activity, could significantly reduce the inflammatory process in the pulmonary exchange system, improving the animal’s ability to breathe and potentially reducing the mortality rate caused by *M. haemolytica* A2.

## 4. Materials and Methods

### 4.1. Bacterium

The *M. haemolytica* A2 strain was a field isolate from a case of pneumonic mannheimiosis obtained from previous works, as was its biochemical and molecular identification [13,35]. For all assays, *M. haemolytica* A2 was grown on blood agar for 24 h and maintained in brain heart infusion (BHI) medium (Dibico, Mexico City, Mexico).

### 4.2. Determination of Subinhibitory Effects of Apo-bLf on the Growth of M. haemolytica A2

To determine the effect of apo-bLf on the growth of *M. haemolytica* A2, incubation with different concentrations of apo-bLf was performed in the culture medium of *M. haemolytica* A2 for 24 h. During this incubation time, culture supernatants (SNs) were obtained, from which the 110-Mh metalloprotease was purified. Lactoferrin was obtained from NutriScience with 97% purity and contained 5 mg of iron per 100 g (NutriScience Innovations, LLC, Milford, CT, USA). Briefly, a pure culture of *M. haemolytica* A2 was performed on blood agar, and a colony was taken and cultured in 5 mL of BHI (Dibico) supplemented with 5, 6, or 8 µM bLf (NutriScience Innovations) (higher concentrations of bLf than in our previous assays), which was maintained for 24 h/200 rpm/37 °C. The initial inoculum comprised 1 × 10^6^ colony-forming units (CFUs). Readings were taken after 9, 12, and 24 h of incubation. Serial dilutions were made, and cultures were plated on a blood agar plate for 24 h at 37 °C to determine the CFUs [36]. Cultures of *M. haemolytica* A2 in BHI (Dibico) without apo-bLf (NutriScience Innovations) were used as the experimental control. All assays were performed in six independent experiments.

### 4.3. Proteases Obtained from Culture Supernatants of M. haemolytica A2 Incubated with apo-bLf

Bacterial growth was performed by removing a colony from a pure culture and transferring it to 5 mL of BHI (Dibico), which was maintained for 24 h at 200 rpm/37 °C. Then, 5% (500 µL) was transferred to 150 mL of BHI (Dibico) and incubated for 24 h at 200 rpm/37 °C with or without two apo-bLf (NutriScience Innovations) concentrations (5 and 6 µM). The culture was subsequently centrifuged at 2600× *g* for 20 min at 4 °C to obtain the SN. The SN was filtered through mixed cellulose ester membranes (Millipore, Dublin, Ireland) of 0.22 μm. Proteins from the filtered SNs were subjected to precipitation with ammonium sulfate (NH_4_)_2_SO_4_ (Sigma-Aldrich, St. Louis, MO, USA) at a saturation percentage of 60%. The samples were centrifuged on 30 kDa cutoff Amicon columns (Millipore) to remove (NH_4_)_2_SO_4_ [37]. Cultures of *M. haemolytica* A2 in BHI (Dibico) without incubation with apo-bLf (NutriScience Innovations) were used as the experimental control. The protein contents of these samples were quantified using the Bradford method [38]. All assays were performed in three independent experiments.

### 4.4. Determination and Identification of 110-Mh Metalloprotease from M. haemolytica A2 and Western Blotting

The 110-Mh metalloprotease was purified using ion exchange chromatography as previously reported [13]. To identify the collagenase, we synthesized a synthetic peptide and subsequently generated antibodies on the basis of the sequence of the 110-Mh metalloprotease (access number: A0A249A431). For this purpose, the bioinformatics programs ABCpred https://webs.iiitd.edu.in/raghava/abcpred/ (accessed on 2 June 2023), BCpred https://webs.iiitd.edu.in/raghava/bcepred/bcepred_submission.html (accessed on 2 June 2023), and kyte-Doolittle https://web.expasy.org/protscale/ (accessed on 2 June 2023), were used, with reference to the sequence near the active site of the protease [39,40,41,42]. The selected peptide (LAHYLEHMILMGSKNYPETNSLDG, residues 104–127) was synthesized. The synthetic peptide was diluted with PBS 1X and emulsified with aluminum hydroxide (Sigma) to increase its immunogenicity. New Zealand female rabbits were subcutaneously and intramuscularly immunized with 0.5 mg of the synthetic peptide. The animals received two immunizations of the same antigenic determinant at the same concentration over an interval of 15 days. Preimmune (PI) serum was obtained before immunization as an experimental control [43].

To corroborate the production of antibodies present in the obtained serum, Western blot assays were performed. After electrophoretic separation by 10% SDS-PAGE, the proteins were transferred to a nitrocellulose membrane (Sigma) for 1 h at 400 mA according to Towbin et al. [44]. The membranes were blocked with skim milk for 2 h at 22 °C and then incubated overnight with rabbit anti-110-Mh metalloprotease serum (1:2000) at 4 °C. The samples were washed three times with PBS-Tween (0.05%) and incubated with an anti-rabbit-HRP secondary antibody (1:3000) (Santa Cruz, Santa Cruz, CA, USA) for 1 h at 22 °C. The membrane was washed seven times, and the signal was developed with a luminol kit (Sigma) using an Odyssey FC imaging system (LI-COR, Lincoln, NE, USA). All assays were performed in three independent experiments.

### 4.5. Effect of Apo-bLf on the 110-Mh Metalloprotease from M. haemolytica A2

After determining that apo-bLf inhibited the secretion of 110-Mh metalloprotease following incubation in *M. haemolytica* A2 culture medium, we evaluated the effect of apo-bLf on the secreted and purified 110-Mh metalloprotease. Zymograms were obtained with 10% acrylamide copolymerized with 0.2% bovine collagen type 1 (Thermo Fisher Scientific, Waltham, MA, USA) as the substrate. Electrophoresis was performed at 4 °C for 2 h at 100 V. The gels were washed for 1 h in 2.5% Triton X-100 (Sigma) and incubated overnight with proteolytic activation buffer at pH 7.0 (100 mM Tris-2 mM CaCl_2_). The gels were stained with 0.5% (*w*/*v*) Coomassie brilliant blue R-250 (Bio-Rad, Feldkirchen, Germany). A proteolytic inhibition assay was performed using 10% SDS-PAGE. Briefly, 10 µg of 110-Mh metalloprotease plus 5 or 6 µM apo-bLf (NutriScience Innovations) with 10 µg of bovine type-I collagen (Thermo Fisher Scientific) was incubated for 12 h at 37 °C in 100 mM Tris–2 mM CaCl_2_ (pH 7.0). After incubation, electrophoresis was performed at a constant voltage (100 V) for 2 h in an ice bath (4 °C). The buffer pH 7 was incubated with bovine type-I collagen (Thermo Fisher Scientific) as the negative control [45]. All assays were performed in three independent experiments.

To quantitatively corroborate the inhibition of *M. haemolytica* A2 collagenase, we performed an inhibition assay with azocoll (Sigma). The azocoll assay is a chromogenic assay based on the binding of an azo group coupled to collagen, and degradation can be quantified via spectrophotometry. Several independent assays were performed on the insoluble fraction (azocoll): 5 µg of 110-Mh metalloprotease, 5 µg of cultured SNs preincubated with 5 or 6 µM apo-bLf (NutriScience Innovations), or 5 µg of 110-Mh metalloprotease with 5 or 6 µM apo-bLf (NutriScience Innovations). Three hundred microliters of 100 mM Tris–2 mM CaCl2, pH 7, was added to these mixtures. To compare the quantification of proteolytic inhibition, the mixture was incubated for 1 h at 22 °C with phenanthroline (Sigma). The mixture was subsequently stirred for 24 h at 37 °C. At the end of the incubation, the samples were subjected to centrifugation at 10,000× *g* for 10 min. The absorbance was measured at a wavelength of 520 nm. The buffer pH 7 was incubated with azocoll (Sigma) as a negative degradation control. Proteolytic activities are reported as milliunits per milligram. This unit of measurement is equivalent to the amount of substrate degraded per minute per milligram of each sample used [46]. All assays were performed in three independent experiments.

### 4.6. Interaction of the 110-Mh Metalloprotease and bLf In Silico

The three-dimensional (3D) structure of the 110-Mh metalloprotease was determined from its sequence, which was obtained from NCBI, because it has not been crystallized. The I-Tasser server https://zhanggroup.org/I-TASSER/ (accessed on 20 May 2023), was used for this purpose. The protein structure was optimized using AutoDockTools and Discovery software with the CHARMm method. Once the 3D structure was obtained, blind docking was performed using the ClusPro 2.0 server, and holo-bLf was used for molecular docking because its 3D structure is available in the Protein Data Bank database https://www.rcsb.org/ (accessed on 20 May 2023), [PDB:1BLf], which was also optimized and similarly pretreated with proteases. The results were analyzed using the AutoDockTools version 1.5.6 (https://autodock.scripps.edu/, accessed on 11 July 2024) and Biovia, Discovery studio 2021 (https://www.3ds.com/products/biovia/discovery-studio/visualization, accessed on 11 July 2024).

### 4.7. Western Blot and Overlay

To corroborate the results of the bioinformatic analysis, an overlay test was performed. For this purpose, 10% SDS-PAGE was performed. Subsequently, 110-Mh metalloprotease was transferred to a nitrocellulose membrane (Sigma) at 400 mA for 1 h, and the membrane was blocked with skim milk for 2 h at 22 °C. The membrane was washed with PBS-Tween (0.05%) and incubated with 1 mg/mL apo-bLf (NutriScience Innovations). The sample was incubated overnight with rabbit anti-bLf (1:2000) antiserum at 4 °C. The samples were washed three times with PBS-Tween (0.05%) and incubated with a secondary anti-rabbit-HRP antibody (1:3000) (Santa Cruz Biotechnology, Santa Cruz, CA, USA) for 1 h at 22 °C. The membrane was washed seven times, and the signal was revealed with a luminol kit reagent (Sigma) using an Odyssey FC imaging system (LI-COR). Preimmune serum previously obtained was used as the experimental control [47]. All assays were performed in three independent experiments.

### 4.8. Statistical Analysis

The experimental design was performed with repeated observations. A comparison of models with regression analysis was used with a significance level of *p* = 0.05. Differences were considered statistically significant when the *p*-value was less than 0.05. Statistical analyses were performed using GraphPad Prism Software 9.2.0 (GraphPad, San Diego, CA, USA).

## 5. Conclusions

Bovine apo-Lf inhibited the activity and secretion of the 110-Mh metalloprotease from *M. haemolytica* A2. Lactoferrin interfered with the active site of the 110-Mh metalloprotease, destabilized this site, and inhibited its proteolytic activity, which reveals a new activity of Lf as a protease inhibitor. Lactoferrin may be an alternative to some antimicrobial treatments for combatting mannheimiosis; however, in vivo studies are necessary to establish a completely effective treatment for mannheimiosis Notably, no resistance to apo-bLf has been reported, and this protein is commercially available and has been approved by the FDA (Silver Spring, MD, USA) and the European Food Safety Authority as a dietary supplement in food products.

## Figures and Tables

**Figure 1 ijms-25-08232-f001:**
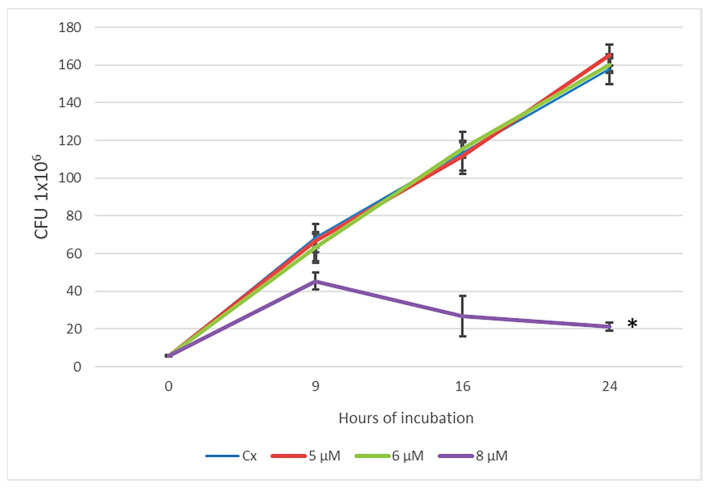
Viability of *M. haemolytica* A2 after 24 h of treatment with different concentrations of apo-bLf. The initial inoculum included 1 × 10^6^ bacteria. Bovine apo-lactoferrin was added to the culture medium, and the bacteria were plated after incubation for the indicated times. CFU readings were recorded. Representative results from six independent experiments are shown. * *p* < 0.05.

**Figure 2 ijms-25-08232-f002:**
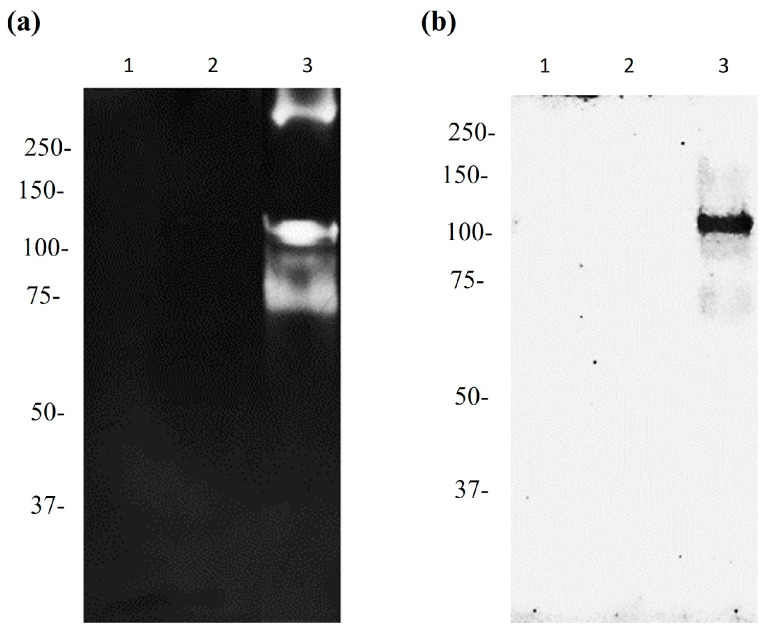
Inhibition of 110-Mh metalloprotease secretion. (**a**) Zymogram copolymerized with bovine collagen type 1. (**b**) Western blot of SNs of *M. haemolytica* A2. The reference molecular weights are shown on the left-hand side. Lanes 1 and 2 correspond to proteins precipitated with 60% ammonium sulfate obtained from *M. haemolytica* A2 culture media supplemented with 5 or 6 µM apo-bLf, respectively. Lane 3 shows the proteolytic degradation pattern without any treatment. Representative results from three independent experiments are shown.

**Figure 3 ijms-25-08232-f003:**
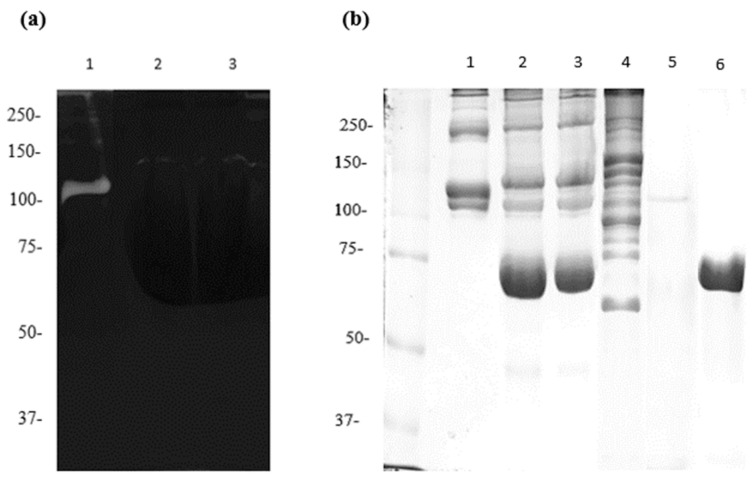
Inhibition of the collagenolytic activity of 110-Mh metalloprotease with subinhibitory concentrations of apo-bLf. (**a**) Zymograms of 10% acrylamide copolymerized with 0.2% bovine collagen type I: 110-Mh metalloprotease without apo-bLf (lane 1) and 110-Mh metalloprotease with 5 or 6 µM apo-bLf (lanes 2 and 3, respectively). (**b**) Ten percent SDS-PAGE was used to evaluate the inhibition of collagen degradation: collagen control (lane 1) and collagen with the 110-Mh metalloprotease with 5 or 6 µM apo-bLf (lanes 2 and 3, respectively). Collagen with the 110-Mh metalloprotease without apo-bLf (lane 4), 110-Mh metalloprotease (lane 5), or apo-bLf alone (lane 6) was used as an experimental control. A representative experiment of three independent samples is shown.

**Figure 4 ijms-25-08232-f004:**
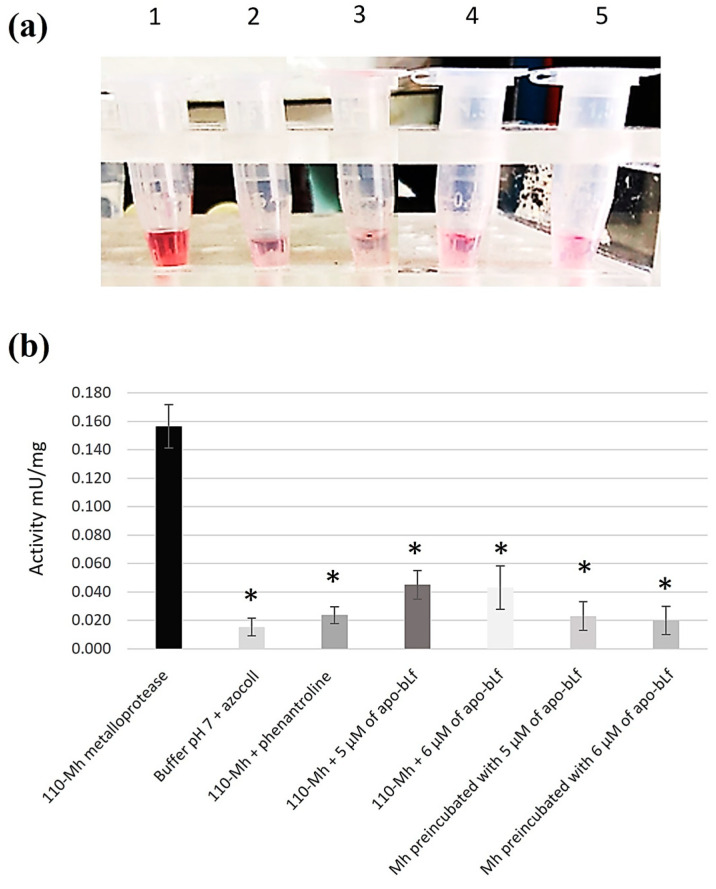
Inhibition of 110-Mh metalloprotease activity in azocoll. The inhibition of the 110 kDa metalloprotease was evaluated with different concentrations of apo-bLf using the azocoll chromogenic assay. (**a**) The 110-Mh metalloprotease without apo-bLf (1), 110-Mh metalloprotease incubated with 5 µM apo-bLf (2), 110-Mh metalloprotease incubated with 6 µM apo-bLf (3), *M. haemolytica* SNs preincubated with 5 µM apo-bLf in the culture medium (4), and *M. haemolytica* SNs preincubated with 6 µM apo-bLf in the culture medium (5). (**b**) Quantification of the proteolytic activity of 110-Mh metalloprotease with or without apo-bLf. Phenanthroline or pH 7 buffer incubated with azocoll were used as experimental controls. Representative results of three independent sample trials are shown. * *p* < 0.05.

**Figure 5 ijms-25-08232-f005:**
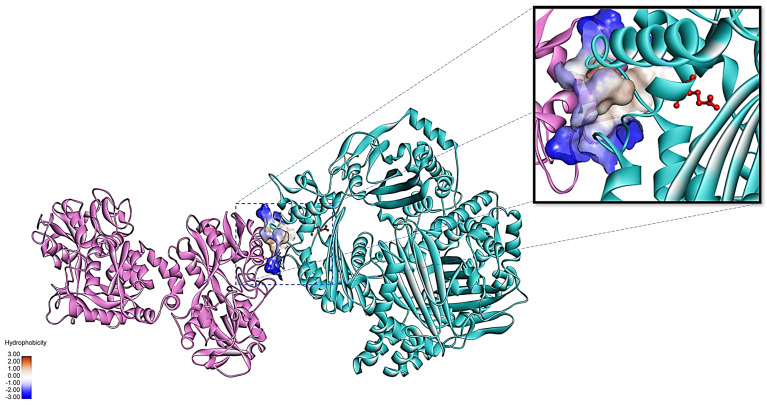
Molecular docking between the bovine lactoferrin model (purple ribbon) and the 110-Mh metalloprotease (blue ribbon). The interaction primarily occurs via hydrophobic interactions. Residue 109 (Glu), where zinc binds within the active site of the 110-Mh metalloprotease, is highlighted in red.

**Figure 6 ijms-25-08232-f006:**
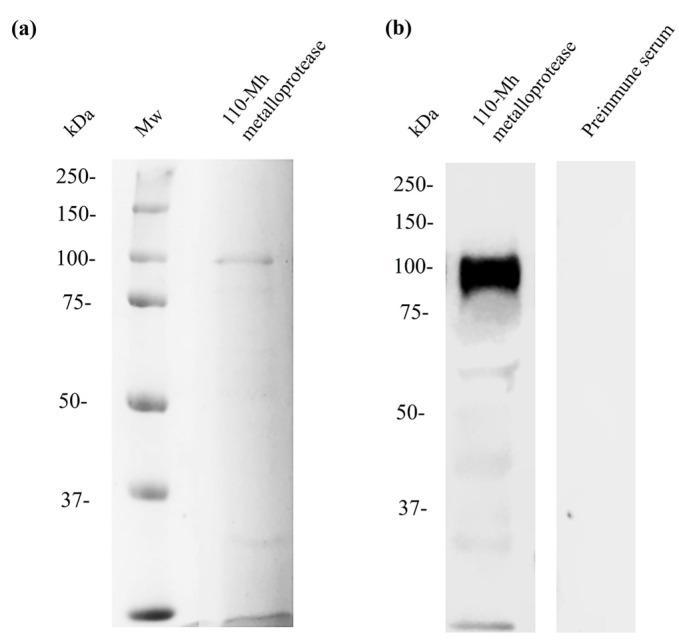
Bovine apo-lactoferrin binding to the 110-Mh metalloprotease using overlay assays. (**a**) Ten percent SDS-PAGE of 110-Mh metalloprotease stained with Coomassie blue. (**b**) The 110-Mh metalloprotease was transferred to a nitrocellulose membrane and incubated with 1 mg/mL apo-bLf. A binding protein of approximately 110 kDa was detected. Representative results of three independent sample trials.

## Data Availability

The original contributions presented in the study are included in the article; further inquiries can be directed to the corresponding author.

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
