# Peer review of "Apo-Lactoferrin Inhibits the Proteolytic Activity of the 110 kDa Zn Metalloprotease Produced by Mannheimia haemolytica A2"

_ijms, 2024, doi:10.3390/ijms25158232_

Round 1
Reviewer 1 Report
Comments and Suggestions for Authors
Introduction.
1) The authors did not make correctly the connection between M. haemolytica and the significance of lactoferrin for its control.
2) There some inaccuracies, for example use of terminology for A2 isolates, lack of efficacy by the vaccine etc. These must be corrected.
3) The objectives of the study must be described clearly.
3) The authors already provide an indication of the advantages of their work compared to previous studies; this must be expanded to give a clear signal of the gaps in the literature that would be filled by publication of this manuscript.
Methodology.
1) Bacterial strains. Please clarify if only one isolate was used in the study.
2) Bacterial strains. If indeed only one isolate was used, then please justify. This is a serious omission to avoid the use of an array of strains and it is not helpful for publishing this work.
3) Control chemicals. Please describe all control chemicals and consumables used in this study.
4) Control procedures. Please explain in detail all control procedures that you followed during this work.
5) Analysis. How many repeat testings were performed? How did you treat the data in relation to analysis?
Results.
1) Tables. Please add tables to present results in summary form.
2) Figures. These are ok.
Discussion.
1) Please explain the clinical consequences of this study.
2) Are the results applicable to other strains of M. haemolytica? If yes, how can we be sure about this?
References.
These are OK.
Conclusions.
1) The conclusions are not fully in line with the findings. I understand that the authors wish to make some extrapolation, but please tone down in the revised version.
Language.
The manuscript requires extensive corrections in English language.
Overall.
Extensive modifications as indicated above and reevaluation for final recommendation.
The use of only one strain is a significant limiting factor for publishing this manuscript.
Comments on the Quality of English Language
Language.
The manuscript requires extensive corrections in English language.
Author Response
"Please see the attachment."

Reviewer 2 Report
Comments and Suggestions for Authors
This study is well-planned and meticulously progresses to confirm a new activity of Lf as a protease inhibitor. The discussion and presentation of data are very detailed. Overall, this study provides insights into the potential of apo-bLf as an antimicrobial therapy and delves deeply into its intervention in pathogenic mechanisms.
Minor issue:
1. In Figure 3, the observed molecular weight of LF appears to be lower than 75 kDa. However, lactoferrin (LF), including apo-lactoferrin (Apo-LF), typically has a molecular weight of around 77-80 kDa. Can you explain the discrepancy between the observed weight in Figure 3 and the expected weight for LF?
Author Response
"Please see the attachment."

Round 2
Reviewer 1 Report
Comments and Suggestions for Authors
No further comments.